# BMP-Induced MicroRNA-101 Expression Regulates Vascular Smooth Muscle Cell Migration

**DOI:** 10.3390/ijms21134764

**Published:** 2020-07-04

**Authors:** Nanju Park, Hara Kang

**Affiliations:** 1Department of Life Sciences, Incheon National University, Incheon 22012, Korea; njp421@naver.com; 2Division of Life Sciences, College of Life Sciences and Bioengineering, Incheon National University, Incheon 22012, Korea; 3Institute for New Drug Development, Incheon National University, Incheon 22012, Korea

**Keywords:** vascular smooth muscle cell, microRNA, miR-101, DOCK4

## Abstract

Proliferation and migration of vascular smooth muscle cells (VSMCs) are implicated in blood vessel development, maintenance of vascular homeostasis, and pathogenesis of vascular disorders. MicroRNAs (miRNAs) mediate the regulation of VSMC functions in response to microenvironmental signals. Because a previous study reported that miR-101, a tumor-suppressive miRNA, is a critical regulator of cell proliferation in vascular disease, we hypothesized that miR-101 controls important cellular processes in VSMCs. The present study aimed to elucidate the effects of miR-101 on VSMC function and its molecular mechanisms. We revealed that miR-101 regulates VSMC proliferation and migration. We showed that miR-101 expression is induced by bone morphogenetic protein (BMP) signaling, and we identified dedicator of cytokinesis 4 (DOCK4) as a novel target of miR-101. Our results suggest that the BMP–miR-101–DOCK4 axis mediates the regulation of VSMC function. Our findings help further the understanding of vascular physiology and pathology.

## 1. Introduction

MicroRNAs (miRNAs) are small noncoding RNAs that regulate a wide range of cellular processes, including development, cell growth, and differentiation. Individual miRNAs generally repress target genes by promoting mRNA destabilization or inhibiting translation [1]. The dysregulation of miRNA expression is associated with various developmental abnormalities and human diseases, including cardiovascular diseases [2]. For example, expression of miR-21 and miR-221 are elevated during vascular neointimal lesion formation following vessel injury, whereas expression of the miR-143/-145 gene cluster is downregulated in the carotid artery after mechanical injury [3,4,5,6].

MicroRNA-101 (miR-101) acts as a tumor suppressor in various cancers. By targeting multiple oncogenes, including *EZH2* (enhancer of zeste homolog 2) and *COX2* (cyclooxygenase-2), miR-101 inhibits cell proliferation, migration, and invasion [7,8,9,10,11,12,13,14,15]. During cancer progression, expression of miR-101 is often suppressed, leading to concomitant upregulation of target gene expression. In glioblastoma, retinoblastoma, prostate, head and neck, bladder, and lung cancers, downregulation of miR-101 causes overexpression of EZH2, a mammalian histone methyltransferase that mediates epigenetic gene silencing, which in turn promotes tumorigenesis. [7,8,9,10,13,15]. COX2 is implicated in cancer development; miR-101 has been reported to modulate COX2 in glioma, colorectal, and lung cancers [11,12,14]. In addition, miR-101 acts as an endogenous proteasome inhibitor by targeting proteasome maturation protein (POMP), a key component of the proteasome biogenesis. The miR-101-mediated interference with proteasome assembly and activity suppresses tumor cell growth [16].

While the importance of miR-101 in cancer progression has been demonstrated, little is known about its functions other than as a tumor suppressor. According to a recent report, miR-101 has antiproliferative effects in vascular cells [17]. IL-6-stimulated activation of JAK/STAT3 signaling promoted cell proliferation in pulmonary microvascular endothelial cells (PMVECs) with hepatopulmonary syndrome. In a rat model of the hepatopulmonary syndrome, miR-101 inhibited the JAK2/STAT3 signaling pathway by targeting JAK2, thereby inhibiting the proliferation of PMVECs. Thus, miR-101 may represent a therapeutic target in vascular disorders, including the hepatopulmonary syndrome.

Vascular smooth muscle cells (VSMCs), which compose the majority of the walls of arterial blood vessels, maintain vessel structures and control blood pressure through contractile and relaxation activities [18]. VSMCs can switch from a contractile phenotype to a proliferative, synthetic phenotype in response to environmental conditions [19]. Differentiated contractile VSMCs lose their quiescence and increase their proliferation and migration during vascular repair or development of vascular pathologies [20]. The phenotypic transition of VSMCs is regulated by several miRNAs, whose aberrant expression can disrupt critical intracellular gene expression networks [2]. For example, bone morphogenetic protein (BMP)-signaling-modulated miRNAs, including miR-21, miR-302, and miR-96, induce differentiation of VSMCs into the contractile phenotype by inhibiting their proliferation and migration [21,22,23]. In contrast, hypoxia-induced miRNAs, such as miR-92b-3p, miR-1260b, miR-497, miR-1268a, and miR-665, promote VSMC proliferation or migration under low oxygen conditions [24,25,26]. However, the involvement of miR-101 in the regulation of VSMC functions remains unclear. Therefore, we aimed to investigate the role of miR-101 in VSMCs and examine the underlying molecular mechanisms.

## 2. Results

### 2.1. miR-101 Affects Functions of VSMCs

To determine whether miR-101 is involved in the regulation of VSMC functions, we first investigated the effects of miR-101 on the migration activity of VSMCs. Pulmonary artery smooth muscle cells (PASMCs) were transfected with negative control miRNA, miR-101 mimic, or antisense inhibitor RNA (anti-miR-101) for 24 h, and then subjected to an *in vitro* scratch wound assay. The migration distance was reduced by 40% in cells transfected with 5 nM miR-101 mimic compared to the migration distance in those transfected with negative control miRNA. (Figure 1A). In contrast, when miR-101 expression was downregulated using anti-miR-101, the cell migration rate was increased by 1.6-fold compared to that of control cells (Figure 1B). These findings indicate that miR-101 inhibits VSMC migration. Since no inhibitor of proliferation was used, effects of miR-101 and/or anti-miR-101 on proliferation cannot be ruled out, but PASMCs migration was inhibited by miR-101 and enhanced by anti-miR-101 as early as 4 h after the scratch was made (Appendix A). The difference in distance of migration between controls seems to be due to the differences in concentrations of control mimic used.

Subsequently, we examined whether miR-101 affects the proliferation of VSMCs. PASMCs were transfected with negative control miRNA, miR-101 mimic, or anti-miR-101 for 72 h and then subjected to a cell proliferation assay using the CellTiter-Glo Luminescent Cell Viability Assay kit. Cell proliferation rate was reduced by 60% in cells transfected with miR-101 mimic (Figure 1C) and enhanced by 3-fold in those transfected with anti-miR-101 (Figure 1D). Thus, miR-101 appears to impair VSMC proliferation. Expression of exogenous miR-101 and knockdown of miR-101 were confirmed by qRT-PCR (Figure 1E,F).

Taken together, these data suggest that miR-101 is likely to have antimigratory and antiproliferative functions in VSMCs. Thus, miR-101 may mimic the effects of BMP signaling, which induces the contractile phenotype of VSMCs.

### 2.2. Expression of miR-101 is Induced by BMP Signaling

As BMP signaling can control the VSMC phenotype through regulating miRNA expression, we investigated whether miR-101 expression is regulated by BMP signaling [21,22,23]. BMP4, which is known to potently induce expression of contractile genes in PASMCs, was used to activate BMP signaling. PASMCs were treated with 6 nM BMP4 for 24 h, before their miR-101 levels were measured by qRT-PCR. BMP4 treatment increased miR-101 levels by approximately 5-fold (Figure 2A). miR-21 and miR-486, the miRNAs whose expression is stimulated by BMP signaling or platelet-derived growth factor (PDGF) signaling, respectively, were included as controls [21,27]. As expected, miR-21 levels were increased, while miR-486 levels were unaffected by BMP4 treatment. These results imply that BMP signaling promotes miR-101 expression.

As PDGF signaling antagonizes BMP signaling and induces a synthetic phenotype in VSMCs, we examined whether miR-101 expression is affected by PDGF signaling. Levels of miR-101, miR-21, and miR-486 were analyzed in PASMCs treated with 40 ng/mL PDGF-BB for 24 h. As expected, miR-486 levels were increased by 3-fold, whereas miR-101 levels were decreased by 88% following the treatment (Figure 2B). These results imply that modulation of miR-101 expression by signaling pathways is important in the regulation of VSMC phenotype and functions. Considering that miR-101 expression is induced by BMP signaling and inhibited by PDGF signaling, miR-101 could be a key mediator of the effects of BMP signaling in VSMCs. Thus, PDGF-induced downregulation of miR-101 may serve to intensify the effects of PDGF signaling.

According to a previous study, miR-101 is upregulated in response to laminar shear stress in vasculature and regulates endothelial cell proliferation [28]. Therefore, our finding that miR-101 expression is induced by BMP signaling in VSMCs suggests the possibility that miR-101 may be involved in regulation of vascular physiology or atherosclerotic diseases associated with VSMC proliferation and migration.

### 2.3. BMP Signaling Induces miR-101 Expression in a Smad-Dependent Manner

To test whether BMP signaling stimulates miR-101 expression through transcriptional regulation, the level of primary miR-101 (pri-miR-101) was measured in PASMCs after 6 h of BMP4 treatment (Figure 3A). Pri-miR-101 levels were increased by 1.5-fold upon BMP4 stimulation, which suggests that the upregulation of miR-101 occurs at the transcriptional level. To test this possibility, PASMCs were treated with actinomycin D (RNA polymerase II inhibitor) prior to BMP4 stimulation. When new transcription was blocked, pri-miR-101 levels were 55% lower than they were in control cells, and no significant increase in pri-miR-101 expression was observed upon BMP4 exposure. These results support the role of BMP signaling in inducing miR-101 expression through transcriptional regulation.

Since BMP-specific signal transducers Smad 1 and Smad 5 are required for the activation of SMC-specific genes by BMP signaling [29], we subsequently examined whether the transcriptional regulation of miR-101 depends on Smad1/5. PASMCs were transfected with negative control siRNA (si-Control) or siRNAs targeting Smad 1 and Smad 5 (si-Smad1/5) for 24 h, before pri-miR-101 levels were measured in the presence or absence of BMP signals using qRT-PCR. Knockdown of Smad 1 and Smad 5 was confirmed by qRT-PCR and immunoblotting (Figure 3B,C). In cells transfected with si-Control, the levels of pri-miR-101 increased by 1.7-fold in response to BMP4 stimulation compared with MOCK control. In contrast, when Smad1/5 was downregulated by siRNAs, the pri-miR-101 levels were reduced by 68% compared with control, and the induction of pri-miR-101 expression by BMP signals was abolished (Figure 3D). The expression level of mature miR-101 was consistent with that of pri-miR-101. BMP4-induced upregulation of miR-101 was dampened in si-Smad1/5-transfected cells (Figure 3E). These results suggest that BMP signaling induces miR-101 expression in a Smad1/5-dependent manner.

To elucidate the mechanisms governing Smad-dependent regulation of miR-101 expression, we mapped the miR-101 promoter region. The putative promoter region was searched through the Ensembl genome browser. Five overlapping regions of the putative miR-101 promoter were cloned into the pGL3-Basic luciferase reporter vector (Figure 4A). The generated luciferase constructs (Luc #1–#5) were transfected into COS7 cells. The luciferase activities of constructs #2, #3, and #4 were 10-, 6.3-, and 9-fold higher than that of the empty reporter vector (Vector), respectively, which implies that the region spanning construct #2 to #4 is involved in promoting miR-101 expression (Figure 4B). In addition, the luciferase activity of construct #2 was increased by BMP4 treatment, while that of the other constructs was unaffected (Figure 4C). These results suggest that the sequence of reporter construct #2 is essential for the BMP-mediated regulation of miR-101 expression.

To investigate whether Smad-dependent regulation occurs through the sequence of reporter construct #2, si-Smad1/5-transfected cells were subjected to a luciferase assay in the presence or absence of BMP signals. Smad1/5 knockdown inhibited the BMP4-induced increase in luciferase activity of construct #2 (Figure 4D). These results suggest that BMP signals regulate miR-101 expression via Smad and provide evidence for a critical role of the sequence of reporter construct #2 in this process. Indeed, computational analysis revealed an evolutionarily conserved Smad-binding element (5′-CAGAC-3′) in the sequence of reporter construct #2 (Figure 4A).

### 2.4. DOCK4 Is a Novel Target of miR-101

To further explore the molecular mechanisms underlying miR-101-mediated regulation of VSMC function, we searched for potential targets of miR-101 using target prediction algorithms, including TargetScan and miRWalk. Interestingly, all members of the dedicator of cytokinesis (DOCK) protein family, except for DOCK3 and DOCK8, were predicted. BMP signaling promotes VSMC contractility by activating miR-21 and, in turn, downregulating the expression of DOCK family proteins [21]. In particular, DOCK4, 5, and 7, as targets of miR-21, are involved in VSMC migratory function. To examine whether DOCK4, 5, and 7 are also targeted by miR-101, PASMCs were transfected with an exogenous miR-101 mimic, and DOCK expression was assessed using qRT-PCR. The mRNA levels of *DOCK4*, but not those of *DOCK5* or *7*, decreased by 50% upon transfection (Figure 5A). When miR-101 activity was inhibited using anti-miR-101, the mRNA levels of endogenous *DOCK4* increased 1.4-fold (Figure 5B). Thus, miR-101 appears to target and destabilize *DOCK4* mRNA. In line with the mRNA results, protein levels of *DOCK4* were reduced by the exogenous miR-101 mimic and enhanced by anti-miR-101 (Figure 5C,D). To further validate that DOCK4 is a novel target of miR-101, we measured the expression of the luciferase reporter construct containing the 3’-UTR of *DOCK4* in the presence of the miR-101 mimic (Figure 5E). The luciferase activity of the 3’-UTR construct was reduced by miR-101, which demonstrates that miR-101 directly targets *DOCK4* via the 3’-UTR. Collectively, these findings imply that the BMP–miR-101 axis regulates VSMC migration by suppressing DOCK4 expression, which in turn may contribute to the inhibition of VSMC migration.

## 3. Discussion

The proliferative and migratory abilities of VSMCs are critical in the regulation of VSMC phenotype and maintenance of vascular homeostasis [19]. miRNAs, which act as post-transcriptional regulators of gene expression, mediate the changes to the VSMC phenotype and contribute to the pathogenesis of various vascular proliferative diseases [2]. In this study, we discovered that miR-101, a well-known tumor-suppressive miRNA, participates in the regulation of VSMC proliferation and migration. We showed that miR-101 expression is induced by BMP signaling in a Smad-dependent manner and that VSMC migration is regulated by the BMP–miR-101 axis through DOCK4 (Figure 5F).

BMP signaling is critical for normal vascular development and homeostasis, and the regulation of miRNAs by BMP signaling is implicated in these processes [30]. BMP signaling stimulates miR-21 production by facilitating the processing of pri-miR-21 to precursor miR-21 through the binding of Smad proteins to a conserved DNA binding sequence (5′-CAGAC-3′) in the miR-21 promoter [31]. Transcriptional regulation of miR-302 by BMP signaling has also been reported [22]. Smad proteins mediate the transcriptional repression of miR-302 by recruiting histone deacetylase to its promoter. Herein, it was shown that BMP signaling upregulated miR-101 expression at the transcriptional level in a Smad-dependent manner via a conserved Smad-binding motif in the promoter region of miR-101. Thus, BMP signaling may regulate VSMC functions by controlling the overall process of miRNA biogenesis through Smad proteins.

We demonstrated that miR-101 inhibits VSMC migration and reduces both the mRNA and protein levels of *DOCK4*. Thus, BMP-induced miR-101 may inhibit VSMC migration by targeting *DOCK4* transcripts. DOCK guanine nucleotide exchange factors regulate cell migration, myogenesis, and clearance of apoptotic cells in mammals [32]. In particular, DOCK4 is a potent regulator of VSMC migration, and its expression is markedly augmented by PDGF signaling [21]. In our study, BMP4 upregulated miR-101, whereas PDGF-BB downregulated miR-101. Since miR-101 suppresses DOCK4 expression, the miR-101–DOCK4 axis may be a critical regulator of VSMC migration.

Emerging evidence suggests that miRNAs act as switches for differentiation and cell fate decisions [33]. BMP and PDGF signaling pathways regulate the transition between the contractile and synthetic phenotype of VSMCs, and their dysregulation can lead to various vascular disorders [19]. As cell migration is a key feature of VSMC phenotype transition, the miR-101–DOCK4 axis may serve as a switch that mediates BMP- and PDGF-induced changes to VSMC phenotype.

## 4. Materials and Methods

### 4.1. Cell Culture

Human primary pulmonary artery smooth muscle cells (PASMCs) were purchased from Lonza (CC-2581, Basel, Switzerland) and were maintained in Sm-GM2 medium (Lonza, CC-3182) containing 5% fetal bovine serum (FBS). Recombinant human BMP4 and PDGF-BB were purchased from R&D Systems (314-BP and 220-BB, Minneapolis, MN, USA). The cells were treated with 6 nM BMP4 or 40 ng/mL PDGF-BB under starvation conditions. For starvation conditions, cells were maintained in Dulbecco’s modified Eagle’s medium (DMEM, SH30243.01) containing 0.2% FBS for 16 h. 

### 4.2. Reverse Transcription Quatitative PCR (qRT-PCR)

Quantitative analysis of the change in expression levels was performed using real-time PCR. The primers used were as follows: pri-miR-101, 5’-GGGGAGCCTTCAGAGAGAGT-3’ and 5’-AGCCACCTGTTTCACATTCC-3’; DOCK4, 5’-TCTGAACTGCTGAAACTTCC-3’ and 5’-GTAGCGGGTAGTATCCTGAA-3’; DOCK5, 5’-AACTCACAGAGCAGCTGAAG-3’ and 5’-TGACTGAGGTGATGGACAAC-3’; DOCK7, 5’-GCAGAACGGTGGCAGCCGAA-3’ and 5’-TCGGTAAGGGGCACTGTGGTGT-3’. The mRNA levels were normalized to 18S rRNA. For quantification of mature miR-101, the miScript PCR assay kit (Qiagen, MS00008379, Hilden, Germany) was used according to the manufacturer’s instructions. Data analysis was performed using a comparative C_T_ method in the Bio-Rad software (Bio-Rad CFX manager 3.1, Hercules, CA, USA). miRNA levels were normalized to U6 small nuclear RNA. The average of three experiments, each performed in triplicate, is presented with standard errors.

### 4.3. miRNA Mimics and Anti-miRNA Oligonucleotides

Chemically modified double-stranded RNAs designed to mimic the endogenous mature miR-101 (5’- CAGUUAUCACAGUGCUGAUGCU -3’) and negative control miRNA were purchased from Genolution (Seoul, Republic of Korea). Antisense inhibitor RNAs (anti-miR-101) were purchased from Bioneer (anti-SMI-002, Daejeon, Republic of Korea). The miRNA mimics and anti-miRNA oligonucleotides were transfected at 5 and 100 nM, respectively, using RNAi Max (Invitrogen, 13778150, Carlsbad, California, CA, USA) according to the manufacturer’s protocol.

### 4.4. RNA Interference

Small interfering RNA (siRNA) duplexes were synthesized by IDT Pharmaceuticals (Coralville, IA, USA). The duplex sequences of siRNAs were as follows: Smad1, 5’-AUAUCAAGAACCUGUUUAGUUUACA-3’ and 5’- UGUAAACUAAACAGGUUCUUGAUAUCA-3’; Smad5, 5’- CAUCAUGAGCUAAAGCCGUUGGATA-3’ and 5’-UAUCCAACGGCUUUAGCUCAUGAUGAC-3’. Negative control siRNA (Genolution, Seoul, Republic of Korea) was used as a control. 

### 4.5. Immunoblotting

Cells were lysed in TNE buffer (50 mM Tris–HCl (pH 7.4), 100 mM NaCl, 0.1 mM EDTA), and total cell lysates were separated by SDS-PAGE, transferred to PVDF membranes, immunoblotted with antibodies, and visualized using an enhanced chemiluminescence detection system (Amersham Biosciences, Little Chalfont, UK). The antibodies used for immunoblotting were an anti-DOCK4 (sc-100718) and an anti-β-actin (sc-47778) from Santa Cruz Biotechnology (Dallas, TX, USA). Smad1 antibody (#6944) and Smad5 antibody (#12534) were purchased from Cell Signaling Technology (Danvers, MA, USA).

### 4.6. Cell Proliferation Assay

CellTiter-Glo Luminescent Cell Viability Assay (Promega, G7572, Madison, WI, USA) was used to determine the number of viable cells in culture. Briefly, 5 × 10^3^ cell/well were seeded in 96-well plates in triplicate. After transfection of miRNAs for 3 days, a volume of CellTiter-Glo reagent equal to the volume of cell culture medium was added to each well. The plates were shaken for 2 min to induce cell lysis and further incubated for 10 min to stabilize the luminescent signal. As there is a linear relationship between the luminescent signal and the number of cells, cell proliferation was measured by reading the absorbance at 490 nm using a GloMax 96 Microplate Luminometer (Promega, Madison, WI, USA). Fold change was calculated as the ratio of recorded luminescence values.

### 4.7. In Vitro Scratch Wound Assay

PASMCs plated in six-well plates were transfected with miR-101 or anti-miR-101, and three scratch wounds were generated with a 1 mL disposable pipette tip. Scratch wounds were photographed over 24 h with a ZEISS inverted microscope with an attached digital camera (ZIESS, Oberkochen, Germany), and their widths were quantitated with ImageJ software (Wayne Rasband, National Institutes of Health, USA). Distance of migration was calculated by subtracting the width measured at a given time from the width initially measured.

### 4.8. Luciferase Reporter Constructs

To map the miR-101 promoter, five overlapped regions of the putative promoter sequence of miR-101 (~5.2 kb) were cloned into the pGL3-Basic vector containing the luciferase gene (Addgene, Watertown, Massachusetts, USA). PCR was used to amplify the promoter sequence of miR-101 from genomic DNA isolated from PASMCs. Primers used are as follows: Luc #1, 5’-AATCTCGAGCAACACCAGGCACAATCTAATG-3’ and 5’-CACACGCGTAGTCCTCTTTTGCCTGCTCA-3’; Luc #2, 5’-CATCTCGAGGCTAACAACAACAAACCCAGTC-3’ and 5’-CATACGCGTCATTAGATTGTGCCTGGTGTTG-3’; Luc #3, 5’-ATTCTCGAGGGTAGCAGAGTGAGGGAAAGAA-3’ and 5’-CATACGCGTGACTGGGTTTGTTGTTGTTAGC-3’; Luc #4, 5’-ACTCTCGAGCAAGTTCAAATAAGCCTGCAAA-3’ and 5’-ATGACGCGTCTTCTCCCTGCCTTCCTGT-3’; Luc #5, 5’-ATGCTCGAGGTATTTTCTGCTCCACTTCCAA-3’ and 5’-ATGACGCGTTTGCAGGCTTATTTGAACTTG-3’.

For luciferase assay of the 3’-UTR of *DOCK4*, the 3’-UTR sequence of *DOCK4* (1215bp) was cloned into the pIS0 vector containing the luciferase gene (Addgene). RT-PCR was used to amplify the 3’-UTR sequence of *DOCK4* from cDNA isolated from PASMCs using 5’-ATGGAGCTCGTCACTTTTCTATGTACCTGCG-3’ and 5’-CTCGGCCGGCCATTTACCATTCAGCAGCAAC-3’ [21].

### 4.9. Luciferase Assay

For the miR-101 promoter analysis, Cos7 cells were transfected with luciferase reporter constructs using Lipofectamine 2000 (Life technologies, Carlsbad, California, USA) for 16 h and then treated with 6 nM BMP4. For the miR-101 target validation, Cos7 cells were cotransfected with 5 nM miR-101 or control mimic and luciferase reporter constructs using Lipofectamine 2000 (Life technologies). A β-galactosidase expression plasmid was used as an internal transfection control. Twenty-four hours later, luciferase assays were performed, and luciferase activity was presented after normalization to β-galactosidase activity.

### 4.10. Statistical Analysis

For each of the assays, three experiments were performed in triplicate, and the results were presented as the average with standard error. Statistical analyses were performed by an analysis of variance followed by Student’s *t*-test using Prism 8 software (GraphPad Software Inc., San Diego, California, USA). *P* values of < 0.05 were considered significant and are indicated with asterisks.

## Figures and Tables

**Figure 1 ijms-21-04764-f001:**
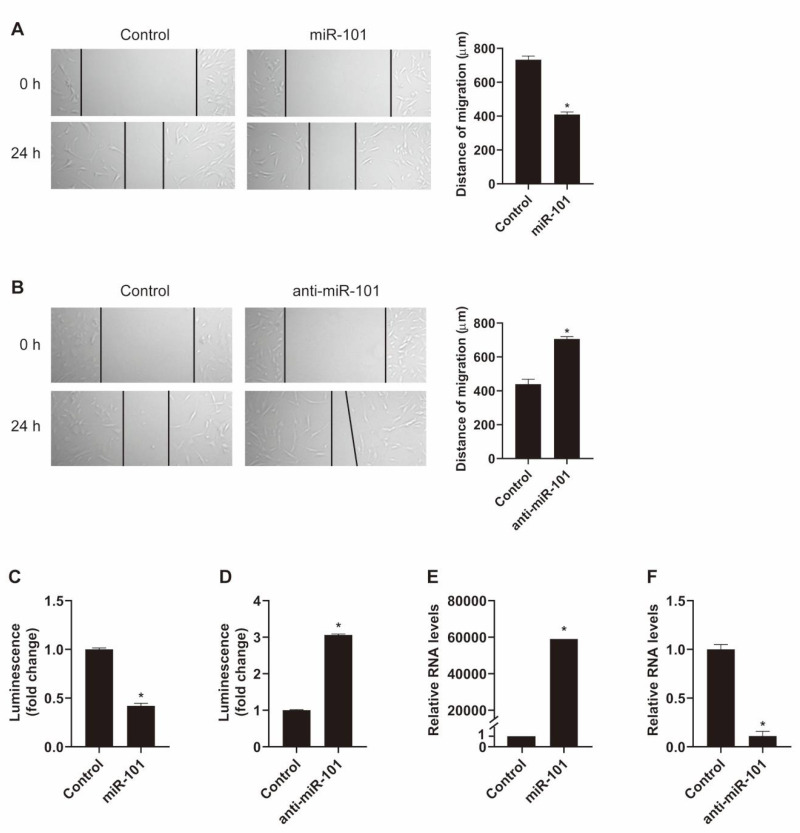
miR-101 regulates vascular smooth muscle cell (VSMC) functions. (**A**,**B**) Pulmonary artery smooth muscle cells (PASMCs) transfected with control, miR-101 mimic (**A**), or anti-miR-101 (**B**) were subjected to the scratch wound assay. The distance of the migration was measured using ImageJ at 24 h after a scratch wound was introduced. The left panel shows representative images, and the right panel shows the quantification graph of the migration distance. The means ± S.E. of triplicate measurements of three independent experiments are shown. *, *p* < 0.05. (**C**,**D**) Luminescent signals for viable PASMCs transfected with control, miR-101 mimic (**C**), or anti-miR-101 (**D**) for 3 days. *, *p* < 0.05. (**E**,**F**) Levels of miR-101 relative to U6 snRNA measured by qRT-PCR in PASMCs 24 h after transfection of control, miR-101 mimic (**E**), or anti-miR-101 (**F**) The data represent the means ± S.E. of triplicates. *, *p* < 0.05.

**Figure 2 ijms-21-04764-f002:**
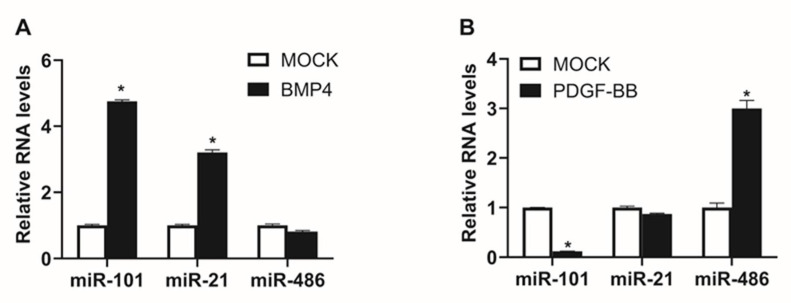
Bone morphogenetic protein (BMP) signaling regulates miR-101 expression. (**A**,**B**) Levels of miRNAs relative to U6 snRNA measured by qRT-PCR in PASMCs 24 h after treatment with BMP4 (**A**) or PDGF-BB (**B**). The data represent the means ± S.E. of triplicates. *, *p* < 0.05.

**Figure 3 ijms-21-04764-f003:**
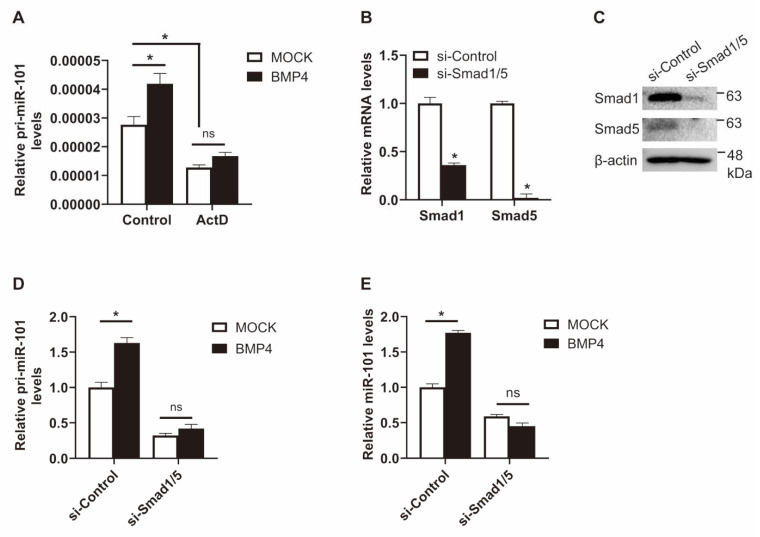
Smads are essential for the regulation of miR-101 expression by BMP signal. (**A**) PASMCs were treated with 25 ng/mL actinomycin D (ActD) for 2 h prior to BMP4 stimulation for 6 h. Levels of pri-miR-101 transcripts relative to 18S rRNA were measured by qRT-PCR. *, *p* < 0.05. (**B**) Levels of Smad1 and Smad5 relative to 18S rRNA in PASMCs transfected with si-Control or siRNAs against Smad1 and Smad5 (si-Smad1/5) were measured by qRT-PCR. *, *p* < 0.05. (**C**) Immunoblot analysis of lysates from PASMCs transfected with si-Control or si-Smad1/5 with anti-Smad1, anti-Smad5, or anti-β-actin antibodies. (**D**,**E**) PASMCs transfected with si-Control or si-Smad1/5 were treated with BMP4. The level of pri-miR-101 transcripts relative to 18S rRNA (**D**) and the level of miR-101 relative to U6 snRNA (**E**) were measured by qRT-PCR. *, *p* < 0.05.

**Figure 4 ijms-21-04764-f004:**
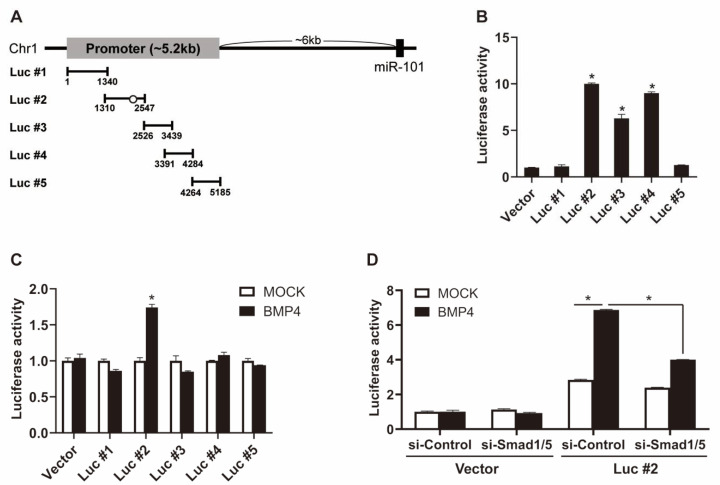
Smad1/5 mediates transcriptional regulation of miR-101. (**A**) Schematic diagram of luciferase reporter constructs containing putative promoter region of the miR-101 gene. A conserved Smad-binding element (5’-CAGAC-3’) is shown as an open circle. (**B**) Luciferase activities of constructs, such as Luc #1~5, were examined in Cos7 cells. An empty vector was used as a control (Vector). The data represent the mean ± S.E. of triplicates. *, *p* < 0.05. (**C**) The relative luciferase activity of each construct with BMP4 stimulation relative to unstimulated is shown. The data represent the mean ± S.E. of triplicates. *, *p* < 0.05. (**D**) Cos7 cells were cotransfected with an empty vector or Luc #2 construct and si-Control or si-Smad1/5 for 24 h, followed by BMP4 treatment for 24 h. Luciferase activities were then measured. The data represent the mean ± S.E. of triplicates. *, *p* < 0.05.

**Figure 5 ijms-21-04764-f005:**
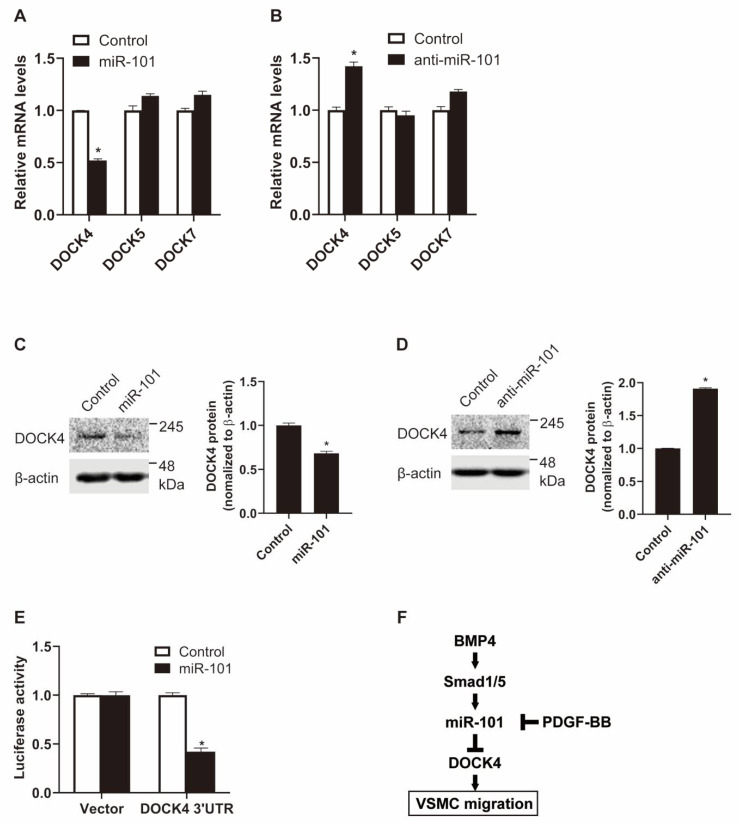
miR-101 targets the 3’-UTR of dedicator of cytokinesis 4 (*DOCK4*) mRNA. (**A**,**B**) Levels of *DOCK4*, *5*, and *7* relative to 18S rRNA in PASMCs transfected with control, miR-101 mimic (**A**), or anti-miR-101 (**B**) were measured by qRT-PCR. *, *p* < 0.05. (**C**,**D**) Total cell lysates from PASMCs transfected with control, miR-101 mimic (**C**), or anti-miR-101 (**D**) were subjected to immunoblot analysis with anti-DOCK4 or anti-β-actin antibodies. By densitometry, relative amounts of DOCK4 protein normalized to β-actin were quantitated. *, *p* < 0.05. (**E**) The luciferase activity of a construct containing the 3’-UTR of *DOCK4* was examined in Cos7 cells by transfecting control or miR-101 mimic. A luciferase vector without 3’-UTR sequence (Vector) was used as a negative control. The data represent the mean ± S.E. of triplicates. *, *p* < 0.05. (**F**) Schematic diagram of the function of miR-101-mediated downregulation of DOCK4 in VSMCs.

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
