# Peer review of "BMP-Induced MicroRNA-101 Expression Regulates Vascular Smooth Muscle Cell Migration"

_ijms, 2020, doi:10.3390/ijms21134764_

Round 1

Reviewer 1 Report

Authors here in this manuscript test their hypothesis about a prominent role of miR-101 in VSMCs and aimed to characterize the mechanism underlying the regulation of VSMCs via miR-101. Their results revealed that miR-101 regulates VSMC proliferation and migration. Further, they showed that miR-101 expression is induced by bone morphogenetic protein (BMP) signaling, and identified dedicator of cytokinesis 4 (DOCK4) as a novel target of miR-101. They conclude that the BMP-miR-101-DOCK4 axis mediates the regulation of VSMC function.

Following suggestions are made for the manuscript under review.

1) Author should show migration and proliferation assay in a DOCK4 complementation assay (by overexpression) in BMP treated PASMCs.
2) Author should show more direct evidence for transcriptional regulation of miR-101expression by SMAD proteins by using some binding assays such as (CHIP or EMSA).
3) Did author verify the role of other SMAD proteins, if no then should mention the reason for selecting SMAD 1/5?

Author Response

Reviewer 1

Authors here in this manuscript test their hypothesis about a prominent role of miR-101 in VSMCs and aimed to characterize the mechanism underlying the regulation of VSMCs via miR-101. Their results revealed that miR-101 regulates VSMC proliferation and migration. Further, they showed that miR-101 expression is induced by bone morphogenetic protein (BMP) signaling, and identified dedicator of cytokinesis 4 (DOCK4) as a novel target of miR-101. They conclude that the BMP-miR-101-DOCK4 axis mediates the regulation of VSMC function.

Following suggestions are made for the manuscript under review.

1) Author should show migration and proliferation assay in a DOCK4 complementation assay (by overexpression) in BMP treated PASMCs.

Response: The role of DOCK4 in PASMCs has already been reported (Kang H, et al., 2012). As DOCK4 is a guanine nucleotide exchange factor for Rac1 GTPase and plays a role in cell migration and cytoskeletal organization, downregulating DOCK4 expression via BMP signaling inhibits PASMC migration. In this study, we reveal that miR-101 regulates PASMC migration, and DOCK4 is a novel target of miR-101.

2) Author should show more direct evidence for transcriptional regulation of miR-101expression by SMAD proteins by using some binding assays such as (CHIP or EMSA).

Response: We agree with the reviewer’s comments. So during a given revision period, ChIP was tried several times to show the physical binding of Smads to miR-101 promoter. However, we could not get the ChIP experiment itself working due to technical issue. It seems that it will take more time to set up the ChIP experiment.

In this study, we showed that BMP4 increases miR-101 expression through transcriptional regulation, and BMP-specific signal transducers Smad 1 and Smad 5 are required for the induction of miR-101 expression by BMP signal (Figure 3). In addition, we identified a promoter site essential for the BMP-mediated induction of miR-101 expression. As a result of the luciferase assay, Smad1/5 knockdown inhibited the BMP-induced increase in luciferase activity of the promoter site that contains Smad-binding element (Figure 4). Therefore, we suggested that BMP signaling induces miR-101 expression in a Smad-dependent manner.

3) Did author verify the role of other SMAD proteins, if no then should mention the reason for selecting SMAD 1/5?

Response: We showed that BMP signaling induces miR-101 expression through transcriptional regulation. Upon BMP signal, the BMP receptor mediates phosphorylation of transcription factors Smad1, 5, and 8, which leads to the subsequent activation of BMP-induced gene transcription in the nucleus (Shi Y, et al., 2003). In a previous study, BMP-specific signal transducers Smad1 and Smad5 are required for the activation of SMC-specific genes by BMP4 (Lagna G, et al., 2007). Therefore, we used Smad1 and 5 to examine whether the transcriptional regulation of miR-101 depends on the transducer of BMP signals.

As suggested by the reviewer, we mentioned the reason for selecting Smad 1/5 in Results section 2.3, as shown below.

“Since BMP-specific signal transducers Smad 1 and Smad 5 are required for the activation of SMC-specific genes by BMP signaling (Lagna G, et al., 2007), we subsequently examined whether the transcriptional regulation of miR-101 depends on Smad1/5.”

Reviewer 2 Report

This article by Park and Kang provides some interesting and novel findings regarding the role of miR-101 in smooth muscle vascular cells. The fact the miR101 is regulated by BMP4 and regulated downstream the activity of Dock4 is valuable to the field of vascular biology.

While the introduction requires significant improvement to ensure the reader understand the importance of the data presented here, the articles and the signaling pathways identified are valuable. However there are also some improvement to be made in the result section and some concern regarding the scratch migration assay that needs to be addressed.

I have listed below the major points that need to be adressed, as well as minor concerns. I am happy to provide clarification if the authors need some.

Major points to be addressed:

  • Introduction: ‘Vascular smooth muscle cells (VSMC), which compose the majority of the walls of blood vessels, maintain vessel structures and control blood pressure through contractile and relaxation activities[14].” The most abundant blood vessels are the capillaries more than 85% of the vascular surface and they do not have a smooth muscle layer. So, this statement needs to be revised. I guess the author refers to arterial, arteriolar, veins and venules.
  • Please edit mistake in reference 14
  • I think that the author might use some of the information provide in the reference 2 to strengthen the rationale for their study as this reference indicated (fig 2) miRNA-101 could be involved in atherosclerotic diseases, which as associated with VSMC proliferation and migration. Authors will need to find the original article that indicates that miR101 is dysregulated during these diseases.
  • When performing the scratch assay it is unclear whether the authors used an inhibitor of proliferation. If not this needs to be specify as the scratch assay will indicate both impact of proliferation and migration collectively. Authors must also adjust their conclusion regarding the results of this assay accordingly if they have not used an inhibitor of proliferation during the assay.
  • Figure 1: Authors need to provide the information regarding the unit (nm, um), readers should not have to guess that it is um.
  • Why are the distance of migration different between the two controls (one presented in fig 1A and one in figure 1B)? I feel the author might want to provide or at least comment on this variability. Is it due to the fact that experiments were done at different passages (older cells), or variability in cell density at start?
  • “Taken together, these results suggest that miR-101 exhibits antimigratory and antiproliferative actions in VSMCs.” Authors might be careful about this statement if they have not used inhibitor of proliferation during scratch assay, since the assay in the absence of this inhibitor measures proliferation and migration together.
  • It will be highly valuable if the authors can provide a rationale for choosing the specific BMP4 protein and not another member of the BMP protein family. This is crucial as the BMP proteins are not discussed in the introduction.

As well as minor points, that can be more easily addressed:

  • Consider revising style in the following phrase “Because the tumor-suppressive miRNA, miR-101, has been suggested to be a critical regulator of cell proliferation in vascular disease, we hypothesized a prominent role of miR-101 in VSMCs and aimed to characterize the mechanism underlying the regulation of VSMCs via miR-101.”
    • ‘has been suggested to be’ please try to use direct voice
    • ‘a prominent role in VSMC’ seems vague ‘a prominent role in controlling …. ??? Maybe ‘Because previous studies reported …., we hypothesize that miR-101 controls important cellular process in VSMC. The present study aimed to identify these processes and the underlying mechanisms characterizing them’??’ I am unsure as the current sentence is vague and unspecific.
  • Introduction:
    • Consider editing first line of introduction: “MicroRNAs (miRNA) are small non-coding RNAs that regulate gene expression in a wide range of cellular processes…” something seems missing and or redundant with the next sentence. So, I am suggesting to first focus on the fact that miRNAs control multiple process. See a suggestion: “MicroRNAs (miRNA) are small non-coding RNAs that regulate in a wide range of cellular processes…”
    • “MicroRNA-101 (miR-101) acts as a tumor suppressor in various cancers. By targeting multiple oncogenes, including EZH2 (enhancer of zeste homolog 2) and COX2 (cyclooxygenase-2), miR-101 inhibits cell proliferation, migration, and invasion.” Add references for these statements in the sentence 3-11.
    • The authors states “Recent reports’ but there is only one reference this suggests that either a reference is missing or that there is just one recent report. Please correct text accordingly.

Author Response

Reviewer 2

This article by Park and Kang provides some interesting and novel findings regarding the role of miR-101 in smooth muscle vascular cells. The fact the miR101 is regulated by BMP4 and regulated downstream the activity of Dock4 is valuable to the field of vascular biology.

While the introduction requires significant improvement to ensure the reader understand the importance of the data presented here, the articles and the signaling pathways identified are valuable. However there are also some improvement to be made in the result section and some concern regarding the scratch migration assay that needs to be addressed.

I have listed below the major points that need to be adressed, as well as minor concerns. I am happy to provide clarification if the authors need some.

Major points to be addressed:

1) Introduction: ‘Vascular smooth muscle cells (VSMC), which compose the majority of the walls of blood vessels, maintain vessel structures and control blood pressure through contractile and relaxation activities[14].” The most abundant blood vessels are the capillaries more than 85% of the vascular surface and they do not have a smooth muscle layer. So, this statement needs to be revised. I guess the author refers to arterial, arteriolar, veins and venules. Please edit mistake in reference 14

Response: We corrected the mistake in the sentence.

“Vascular smooth muscle cells (VSMC), which compose the majority of the walls of arterial blood vessels, maintain vessel structures and control blood pressure through contractile and relaxation activities [14].”

2) I think that the author might use some of the information provide in the reference 2 to strengthen the rationale for their study as this reference indicated (fig 2) miRNA-101 could be involved in atherosclerotic diseases, which as associated with VSMC proliferation and migration. Authors will need to find the original article that indicates that miR101 is dysregulated during these diseases.

Response: We agree with the reviewer’s comments, and have added more information and explanation to our Introduction, and Results section 2.2.

(Introduction)

“For example, expression of miR-21 and miR-221 are elevated during vascular neointimal lesion formation following vessel injury, whereas expression of the miR-143/-145 gene cluster is downregulated in the carotid artery after mechanical injury.”

(Results 2.2)

“According to a previous study, miR-101 is upregulated in response to laminar shear stress in vasculature, and regulates endothelial cell proliferation [28]. Therefore, our finding that miR-101 expression is induced by BMP signaling in VSMCs suggests the possibility that miR-101 may be involved in regulation of vascular physiology or atherosclerotic diseases associated with VSMC proliferation and migration.”

3) When performing the scratch assay it is unclear whether the authors used an inhibitor of proliferation. If not this needs to be specify as the scratch assay will indicate both impact of proliferation and migration collectively. Authors must also adjust their conclusion regarding the results of this assay accordingly if they have not used an inhibitor of proliferation during the assay.

Response: We agree with the reviewer’s comments. Since no inhibitor of proliferation was used, the scratch wound assay could show the effect of miR-101 on proliferation and migration abilities collectively. In fact, when we conducted the scratch wound assay, we also took pictures at early timepoints and analyzed the distance of migration. Migration was inhibited in PASMCs transfected with miR-101, but enhanced in PASMCs transfected with anti-miR-101 compared with control even at 4 h and 8 h after the scratch was generated. Therefore, we thought that miR-101 inhibits cell proliferation as well as migration. The data is now added to Supplementary Figure 1. We also mentioned this in Results 2.1, as shown below.

“Since no inhibitor of proliferation was used, effects of miR-101 and/or anti-miR-101 on proliferation cannot be ruled out, but PASMC migration was inhibited by miR-101, and enhanced by anti-miR-101 as early as 4 h after the scratch was made.”

4) Figure 1: Authors need to provide the information regarding the unit (nm, um), readers should not have to guess that it is um.

Response: The analysis function of ImageJ was used to measure the length (distance of migration). The real measurement unit is μm. The unit is now indicated on the Y axis of the graphs in Figure 1.

5) Why are the distance of migration different between the two controls (one presented in fig 1A and one in figure 1B)? I feel the author might want to provide or at least comment on this variability. Is it due to the fact that experiments were done at different passages (older cells), or variability in cell density at start?

Response: We tried to perform all experiments using cells of the same passage number and cell density, to the extent this is possible. I conjecture that cell migratory activities are affected by the concentrations of control mimic used. For the overexpression experiment, 5 nM of control and miR-101 mimic were used; for the knockdown experiment, 100 nM of control and anti-miR-101 were used. We often see this phenomenon.

As suggested by the reviewer, we mentioned this below in Results 2.1.

“The difference in distance of migration between controls seems to be due to differences in concentrations of control mimic used.”

6) “Taken together, these results suggest that miR-101 exhibits antimigratory and antiproliferative actions in VSMCs.” Authors might be careful about this statement if they have not used inhibitor of proliferation during scratch assay, since the assay in the absence of this inhibitor measures proliferation and migration together.

Response: We agree with the reviewer’s comments. As in answer to the third comment, migration was inhibited in PASMCs transfected with miR-101, and enhanced in PASMCs transfected with anti-miR-101 compared with control even at 4 h and 8 h after the scratch was generated. Therefore, we thought that miR-101 inhibits cell proliferation as well as migration.

We corrected the sentence.

“Taken together, these data suggest that miR-101 is likely to have antimigratory and antiproliferative functions in VSMCs.”

7) It will be highly valuable if the authors can provide a rationale for choosing the specific BMP4 protein and not another member of the BMP protein family. This is crucial as the BMP proteins are not discussed in the introduction.

Response: BMPs have been reported to inhibit proliferation and induce apoptosis in PASMCs (Morrell N, et al., 2001 and Zhang S, et al., 2003). In human PASMCs, BMP ligands BMP2-6, but not BMP7, are expressed (Zhang S, et al., 2003 and Lagna G, et al., 2007). Among those expressed, BMP4 induces SMC-specific gene expression most potently (Lagna G, et al., 2007). Therefore, BMP4 was chosen for our experiments to investigate whether BMP signaling regulates miR-101 expression.

As suggested by the reviewer, we now mention this in Results section 2.2, as shown below.

“BMP4, which is known to most potently induce expression of contractile genes in PASMCs, was used to activate BMP signaling.” 

As well as minor points, that can be more easily addressed:

  1. Consider revising style in the following phrase “Because the tumor-suppressive miRNA, miR-101, has been suggested to be a critical regulator of cell proliferation in vascular disease, we hypothesized a prominent role of miR-101 in VSMCs and aimed to characterize the mechanism underlying the regulation of VSMCs via miR-101.”

‘has been suggested to be’ please try to use direct voice

‘a prominent role in VSMC’ seems vague ‘a prominent role in controlling …. ??? Maybe ‘Because previous studies reported …., we hypothesize that miR-101 controls important cellular process in VSMC. The present study aimed to identify these processes and the underlying mechanisms characterizing them’??’ I am unsure as the current sentence is vague and unspecific.

Response: As suggested by the reviewer, the sentences were modified more clearly, as shown below.

“Because a previous study reported that the tumor suppressive miRNA, miR-101, is a critical regulator of cell proliferation in vascular disease, we hypothesized that miR-101 controls important cellular processes in VSMCs. The present study aimed to elucidate the effects of miR-101 on VSMC function, and its molecular mechanisms.”  

2) Introduction:

Consider editing first line of introduction: “MicroRNAs (miRNA) are small non-coding RNAs that regulate gene expression in a wide range of cellular processes…” something seems missing and or redundant with the next sentence. So, I am suggesting to first focus on the fact that miRNAs control multiple process. See a suggestion: “MicroRNAs (miRNA) are small non-coding RNAs that regulate in a wide range of cellular processes…”

Response: As suggested by the reviewer, we corrected the first sentence of Introduction, as shown below.

“MicroRNAs (miRNA) are small non-coding RNAs that regulate a wide range of cellular processes, including development, cell growth, and differentiation.”

3) “MicroRNA-101 (miR-101) acts as a tumor suppressor in various cancers. By targeting multiple oncogenes, including EZH2 (enhancer of zeste homolog 2) and COX2 (cyclooxygenase-2), miR-101 inhibits cell proliferation, migration, and invasion.” Add references for these statements in the sentence 3-11.

Response: As suggested by the reviewer, we added the references to the revised manuscript.

4) The authors states “Recent reports’ but there is only one reference this suggests that either a reference is missing or that there is just one recent report. Please correct text accordingly.

Response: It was corrected, as shown below.
“According to a recent report, miR-101 has antiproliferative effects in vascular cells.”

Round 2

Reviewer 2 Report

I would like to thank the authors for addressing all my comments.

The supplemental figure is valuable and the editing performed ease reading.